# Successful and Unsuccessful Brain Aging in Pets: Pathophysiological Mechanisms behind Clinical Signs and Potential Benefits from Palmitoylethanolamide Nutritional Intervention

**DOI:** 10.3390/ani11092584

**Published:** 2021-09-03

**Authors:** Caterina Scuderi, Lorenzo Golini

**Affiliations:** 1Department of Physiology and Pharmacology “Vittorio Erspamer”, Sapienza University of Rome, P.le A. Moro, 5, 00185 Rome, Italy; 2Neurology and Neurosurgery Service, Department of Small Animal Surgery, Vetsuisse Faculty, University of Zurich, Winterthurerstrasse 260, 8057 Zurich, Switzerland; lorenzo.golini@uzh.ch

**Keywords:** aging, cognitive dysfunction syndrome, neuroinflammation, neurodegeneration, astrocyte, microglia, mast cells, cat, dog, palmitoylethanolamide

## Abstract

**Simple Summary:**

Cognitive dysfunction syndrome is a common yet underreported neurodegenerative disorder of elderly dogs and cats and a natural model of human Alzheimer’s disease. The increasingly expanding life expectancy means a larger proportion of affected animals in the coming decades. Although far from being curative, available treatments are more effective the sooner they are started. Educating veterinary practitioners and owners in the early recognition of age-related cognitive dysfunction is thus mandatory. By shedding light on the mechanism underlying the disease, novel and more effective approaches might be developed. Emerging evidence shows that successful and unsuccessful brain aging share a common underlying mechanism that is neuroinflammation. This process involves astrocytes, microglia, and mast cells and has a restorative homeostatic intent. However, for reasons not fully elucidated yet, neuroinflammation can also exert detrimental consequences substantially contributing to neurodegeneration. Here we summarize the evidence accumulated so far on the pathogenic role of neuroinflammation in the onset and progression of age-related neurodegenerative disorders, such as Alzheimer’s disease. The potential benefit of palmitoylethanolamide dietary intervention in rebalancing neuroinflammation and exerting neuroprotection is also discussed.

**Abstract:**

Canine and feline cognitive dysfunction syndrome is a common neurodegenerative disorder of old age and a natural model of human Alzheimer’s disease. With the unavoidable expanding life expectancy, an increasing number of small animals will be affected. Although there is no cure, early detection and intervention are vitally important to delay cognitive decline. Knowledge of cellular and molecular mechanisms underlying disease onset and progression is an equally decisive factor for developing effective approaches. Uncontrolled neuroinflammation, orchestrated in the central nervous system mainly by astrocytes, microglia, and resident mast cells, is currently acknowledged as a hallmark of neurodegeneration. This has prompted scientists to find a way to rebalance the altered crosstalk between these cells. In this context, great emphasis has been given to the role played by the expanded endocannabinoid system, i.e., endocannabinoidome, because of its prominent role in physiological and pathological neuroinflammation. Within the endocannabinoidome, great attention has been paid to palmitoylethanolamide due to its safe and pro-homeostatic effects. The availability of new ultramicronized formulations highly improved the oral bioavailability of palmitoylethanolamide, paving the way to its dietary use. Ultramicronized palmitoylethanolamide has been repeatedly tested in animal models of age-related neurodegeneration with promising results. Data accumulated so far suggest that supplementation with ultramicronized palmitoylethanolamide helps to accomplish successful brain aging.

## 1. Successful Aging and the Brain

Over the last decades, life expectancy is getting longer, and the traditional 65-year chronological age threshold for entering old age has recently been questioned in favor of the fixed remaining life expectancy as a better indicator [1]. Similarly, the life expectancy of dogs and cats has been increasingly expanding during the last years, mainly thanks to the tremendous improvement of veterinary nutrition and medicine [2,3,4]. Currently, dogs are considered senior when they enter the last 25% of their predicted life expectancy, while feline pets are viewed as “old” between 10 and 14 years, with geriatric age beginning thereafter [3].

In human medicine, successful aging is a multidimensional term to describe seniors with minimal physical and health impairment, autonomous and satisfactory social interactive life, as well as prompt cognitive function [5]. In veterinary medicine, “successfully aging” pets are considered those who do not show important impairments in daily routine [6] while usually manifesting slow deterioration in general activity and playing behavior [7]. Failure to respond to commands, decreased interest toward novelty, increased time spent sleeping, and frequency of phobias, as well as reduced ability to cope with mild social challenges are further features of normal aging in pets [8,9,10,11]. Learning and memory are not negatively affected in most cases, although cognitive abilities tend to slow down over time [12,13], especially if comorbidities are present, due to the recently acknowledged “frailty syndrome” of the elderly [14,15].

## 2. Unsuccessful Brain Aging in Pets

Laboratory beagle dogs have been involved in aging studies since the mid-1990s, and behavioral changes compatible with pathological aging were indeed identified [16]. However, the laboratory aseptic conditions and requirement for pre-test training lowered the possibility to apply the same lab tests to client-owned pets [17], thus hampering our understanding of age-related changes in the complex field of inter- and intra-specific social interactions, as well as the effect of environment on the development of pathological aging [9,18,19]. Nonetheless, laboratory beagles have been the first dog breed in which neuropathological changes similar to those observed in human beings suffering from sporadic Alzheimer’s disease (AD) were reported [20]. Neuropathological studies performed in dogs of this breed have indeed confirmed the relationship between β-amyloid load and cognitive decline in otherwise neurological normal dogs [20,21], as will be further discussed later.

In the late 1990s, age-related cognitive decline was first described in client-owned dogs and cats, as recently reviewed [19,22]. Since then, age-related cognitive decline in pets has been a hot topic in veterinary medicine, mainly due to its striking similarities with human AD, as will be discussed further in this review. The canine counterpart of senile dementia of Alzheimer’s type (ccSDAT) [23] and canine cognitive dysfunction syndrome or CCDS [24] are just a couple of several different terms that have been coined to refer to pathological brain aging. According to Patrick Pageat, age-related cognitive impairment is to be considered a clinical umbrella disorder and should be further divided into four entities: hyperaggressiveness in old dogs; confusional syndrome of old dogs; involutive depression; and dysthymia [25]. Unsuccessful brain aging with AD-like cognitive decline in elderly pets will be hereafter referred to as cognitive dysfunction syndrome (CDS), in order to denote a progressive age-related disorder affecting either dogs or cats.

## 3. Neurobehavioral and Physical Signs

CDS encompasses behavioral (i.e., neuropsychological) [26,27,28] and physical signs [18,29]. The neuropsychological signs of canine and feline CDS are well described and classically summarized with the acronym D.I.S.H.A. that stands for Disorientation, altered Interactions, Sleep-wake cycle changes, breaking in the House soiling and altered Activity levels with newly onset anxiety [27,28,30,31,32]. Disorientation, or confusional mental status, is a common finding (over half of the cases), with affected dogs mainly showing aimless stereotypic wandering or stopping in front of the wrong edge of the front door. Vocalization and staring at the empty space as well as getting “stuck” in corners or narrow places are further frequent signs (Figure 1).

Altered interaction with owners is reported in about 50% of CDS dogs, and increased aggressive behavior can also be observed, albeit less frequently [33]. Sleep-wake cycle disturbance is one of the commonest clinical features and is characterized by an extremely altered REM cycle with frequent awakening during the night and increased daytime sleep [34]. Lost housetraining has been reported in about one-third of CDS patients, and in 16%, active soiling could be observed. Altered emotions with increased novelty anxiety are also frequently detected [27,28,31]. The main neurobehavioral signs of canine CDS are summarized in Table 1.

A very interesting clinical parallelism with AD in human beings is the recognition of an intermediate phase between clinical normal aging subjects and clinically demented pets. This intermediate step has also been named mild cognitive impairment (MCI), borrowing it from human medicine [30,31]. MCI has been defined as reduced cognitive ability without significant interference with daily life [35]. A recent study showed that dogs with MCI exhibit reduced social interaction and novelty fear without signs of breaking of the housetraining nor sleep–awake cycle disturbance [31].

On the physical side of CDS, a large web-based survey recently performed in dogs aged 10 years or older found that vision impairment, smell disturbance, tremor, swaying or falling, and head ptosis were the main physical disturbances related to CDS [29]. Interestingly, similar signs are also prevalent in demented human patients [36] and extrapyramidal signs within an MCI state are predictive of faster progression to full dementia [37]. In this context, it is noteworthy that dogs with CDS were found to be twice as likely to show neurologic deficits compared to “successfully aging” dogs [18].

All that said, CDS is currently considered to be a natural model of human AD [29,38,39,40] because neuropathological changes closely parallel those of the AD-affected brain [41,42], as will be further discussed in the following paragraphs.

## 4. Diagnostic Considerations

First and foremost, it is important to note that CDS can be definitively diagnosed only after death (with microscopic examination of the brain). The presumptive diagnosis of CDS is based on the pet’s medical history and recognition of behavioral signs. Moreover, the exclusion of other medical causes that might mimic CDS must be carefully considered [43].

Several questionnaires have been developed to support CDS diagnosis [24,30,44]. Some of them have been clinically linked to amyloid brain deposition [23,45], lactate and pyruvate concentration [46], and, more recently, neuroinflammation [31]. Among them, the Canine Cognitive Dysfunction Rating Scale (CCDRS) has an overall diagnostic accuracy of 98.9%, with a negative predictive value of 99.3% and a positive predictive value of 77.8% [30].

Some disorders share similar behavioral signs with CDS and must therefore be included in the differential diagnosis. For example, sudden onset of behavioral alteration, with depression, confusion and disorientation could be related to primary [47] or secondary brain tumors [48]. Moreover, signs of dementia with increased irritability and aggressive behavior might be related to subclinical congenital portosystemic shunts in elderly dogs [49]. Periodontal disease during aging should also be considered since this painful disorder is associated with an increased likelihood of exhibiting cognitive dysfunction signs [50].

### 4.1. Brain Imaging

Although in veterinary medicine evidence, imaging research is scanty compared to the human field, magnetic resonance imaging (MRI) has been recently applied not only to rule out other causes (e.g., brain tumors) but also to gain insight into CDS-associated brain changes. Canine normal brain aging is associated with the volumetric reduction of the frontal cortex, basal ganglia, and hippocampus [51], resulting from the progressive loss of white matter and the subsequent expansion of the ventricular system [52,53]. MRI studies have highlighted that brains from “successfully aging” dogs exhibit less generalized brain atrophy, while brains from CDS animals show more severe and progressive forebrain atrophy paralleling the degree of cognitive impairment [54]. More recently, MRI investigations in dogs confirmed that forebrain atrophy is mainly due to white matter reduction [55,56], decreased hippocampal volume [57], and reduced interthalamic adhesion [58,59] (Figure 2). In particular, an interthalamic adhesion size of 3.82 ± 0.79 mm has been associated with CDS [58]. Increased microhemorrhages [59] and higher signal intensity of the white matter or leukoaraiosis [60] have also been reported in brains from pets affected by CDS.

### 4.2. Putative Biomarkers

Routine blood tests in elderly dogs would show a mild increase of liver enzymes [31], with no other salient abnormalities being definitely confirmed to date. In canine patients with CDS, detection and quantification of soluble oligomers of β-amyloid fragments 1–40 and 1–42, hyperphosphorylated tau protein, and neurofilament light chain are considered promising, although not yet validated, blood biomarkers [31,32,61,62,63,64,65].

## 5. Prevalence and Disease Progression

Although questionnaires have the inherent ability to grade clinical signs, with the resulting scores being a useful method to monitor CDS progression, single itemized clinical signs within the D.I.S.H.A. five categories were originally used [66,67]. MCI was then diagnosed if clinical signs fell under a single category, while signs within two or more categories were indicative of CDS [66]. Using this two-level diagnostic classification, MCI was detected in 13% to 28% of dogs aged 8–12 years [67] as well as 41% to 68% of dogs 13 years old and over [24,66,68]. On the other hand, dogs diagnosed with CDS ranged from 10% (up to 12 years of age) to 61% (13 years old and over) [24,69], with Osella [68] reporting that 33% of the evaluated elderly dog population was affected by this syndrome. It must be mentioned, however, that different authors reported lower data, with cumulative (MCI and CDS) prevalence ranging from 14% [29,70] to 22.5% [71].

Even deterioration rates varied across different studies. According to some authors, normal aging deteriorated to MCI during a four-year period in 58% of the studied dogs [28], while others reported that 42% and 71% of dogs moved from successful aging to MCI in 6 and 12 months, respectively [24]. Conversely, progression from MCI to CDS was detected in 14% of dogs in 4 years [28], 24% in 6 months, and 50% in 12 months [24], depending on the study.

While CDS has long been described (although with different terminology) and frequently seen in older pets, veterinary general practitioners (GPs) are still not accustomed to the diagnosis and management of this progressive and commonly devastating syndrome. This clearly emerges from Australian and North American surveys, where CDS was diagnosed or suspected by local GPs in a percentage of elderly pets as low as 1.9% and 12%, respectively [70,72], confirming CDS to be importantly underdiagnosed.

These findings highlight the need for veterinarians’ and owners’ effective education about age-related cognitive dysfunction. Early recognition of age-related behavioral signs will allow early intervention, which is a critical determinant in chronic progressive disorders, such as CDS [73]. This is especially true if one considers that MCI to CDS progression is fast and that available treatments—although far from being curative—are more effective the sooner they are started [33,74].

## 6. Pathological Features

Canine and feline CDS has been historically associated with the deposition of soluble amyloid fragments in the extracellular compartment and the formation of phosphorylated tangles of tau protein within the axons of the central nervous system (CNS) [21,38,39,42,75,76,77].

On the latter side, several difficulties have been encountered in staining neurofibrillary tangles in the pet brain and correlating their presence with cognitive impairment [19,78]. Only recently, immunohistochemical findings have shown elderly dogs to be indeed affected by microfilament pathology (S396 p-tau) and, albeit rarely, mature neurofibrillary tangles [76,79].

Conversely, old dogs and cats have been repetitively proven to be a good natural model to study the degree and severity of amyloid deposition [19,21]. In the aging canine brain, amyloid deposition seems to follow a distinct progression pathway [80], with a pattern that is similar to the one depicted by Braak and Braak in the brain from human patients affected by AD [75,81]. Beta-amyloid is present as diffuse plaques in 95% of dogs between 10 and 17 years, and its deposition follows four stages [82]. Stage I is characterized by a roundish faint delineated accumulation of small diffuse plaques within cortical layers IV and V. These plaques are positive to immunohistochemistry, but thioflavin- and Congo red-negative and small filaments near the neuronal plasma membrane can be observed [83]. Stage II is characterized by more diffuse deposits within deep cortical layers (V and VI) that tend to invade more superficial areas and become coalescent. In stage III, plaques are dense and invade cortical layers from I to III. Finally, stage IV plaques are dense and Congo-positive. This latter is very rarely observed in dogs. Canine studies have shown that at 9 years of age, stage I involves just 5% of the frontal cortex [80]. In the following 6 months, plaque deposition tends to increase slowly in a craniocaudal fashion and progressively affects the occipital, entorhinal, and parietal cortex.

Pathological changes in the entorhinal cortex and limbic system of aged dogs have been extensively investigated in the last few years [79,84]. In brains from dogs 10 to 15 years old, 22%, 40%, 22.7%, and 4.6% of the limbic system show plaques on stages I, II, III, and IV, respectively [79]. These plaques tend to mature and become denser as the dog ages and cognition declines [45,80].

Although observed in normal canine brain aging [51], vascular β-amyloid deposition has also been associated with clinical signs of CDS and is considered to be a causative factor of the microhemorrhages observed in MRIs [45,85,86].

As in the human brain, neurotoxicity from β-amyloid in pet brains results from several molecular and cellular pathways, comprising changes in ion channels, synaptic loss, decreased neurotrophic factors, as well as impaired mitochondrial redox balance [21,26,87,88,89,90].

One of the most recently and intensively investigated areas in human AD, as well as AD-like disorders of dogs and cats, is neuroinflammation. As detailed below, a vicious cycle between β-amyloid and aberrant tau phosphorylation and activation of non-neuronal cells (e.g., microglia and mast cells) is currently considered the *“primum movens”* of those brain alterations, ultimately resulting in the clinical signs observed in elderly dogs and cats affected with CDS.

## 7. The Neuroinflammatory Process and Its Role in Healthy and Pathological Aging

### 7.1. Neuroinflammation

There is growing interest in understanding the role of neuroinflammation within the brain during both healthy and pathological aging [91]. Indeed, inflammatory processes have been found to be involved in the etiopathology of many neurological disorders. This has led neuroscientists to actively study molecular mechanisms. Therefore, details of immune responses have been at least in part clarified, and additional neuroimmunological disorders have been defined, as well as roles for the immune system have been highlighted in common non-immunological disorders, such as AD [92].

Neuroinflammation is defined as an inflammatory response within the brain or spinal cord [93]. This immune response involves local elements and elements that are transported from the periphery. The local CNS immune system is composed of resident glial cells (mainly microglia and astrocytes), mast cells, and endothelial cells [94]. Glial cells represent an extremely heterogeneous population of cells of neural (astroglia) and non-neural (microglia) origin populating the brain. The heterogeneity of glia correlates with the multiplicity of functions that they perform [95]. All types of glial cells sustain nervous tissue and maintain the function of the CNS as an organ. Together with CNS mast cells, glial cells finely regulate a huge number of central responses (Table 2). Therefore, the interest in the involvement of glia and mast cells in the pathophysiology of AD has grown very much in the last decades [96].

Glial cells, and in particular astrocytes, are also endowed with the ability to rigorously regulate the transport of immune cells and cytokines from the periphery into the brain [116]. Any impairments in this transport system can lead to the activation of brain-resident T cells and B cells (as a result of primary brain processes or influx from the periphery) and cause diseases.

Whatever the triggering cause, neuroinflammation is mediated by the production of cytokines, chemokines, reactive oxygen species, reactive nitrogen species, and many other mediators [117]. These molecules are produced by microglia, astrocytes, mast cells, endothelial cells, and peripherally derived immune cells. The release of these mediators has immune, physiological, and biochemical consequences, which influence most neurological disorders, even those for which the primary etiology is not inflammatory [116]. Further, the degree and extent of neuroinflammation depend on the context, duration, and course of the primary stimulus or insult.

Neuroinflammation is considered a highly preserved physiological defensive process triggered by various insults that can reach the brain and are aimed at restoring homeostasis [118]. It allows the elimination of the underlying cause, repair injured tissue, and restore normal functions. However, under some circumstances, neuroinflammation can also have detrimental consequences. Due to causes not yet fully understood, this acute physiological and pro-resolving process becomes persistent and not-resolving, thus exerting detrimental consequences. To summarize, acute and persistent inflammation show contrasting profiles [119]. Physiologically, neuroinflammation comprises an acute pro-inflammatory phase, in which greater production of pro-inflammatory mediators is initiated and killing functions predominate, and a secondary anti-inflammatory phase, where recovery occurs. When this does not happen, and neuroinflammation is not properly terminated, it turns into a chronic non-resolving process, resulting in brain dysfunction [120] (Figure 3).

The presence of chronic non-resolving neuroinflammation has been reported during aging and suggested as a causative agent and main trigger of the damage associated with many neuroinflammatory and neurodegenerative disorders, such as multiple sclerosis and AD [121,122]. Although many aspects remain to be clarified, scientists agree on the key roles that neuroinflammation plays in brain aging during physiological and neurological conditions.

### 7.2. Neuroinflammation in Alzheimer’s Disease

Accumulated evidence suggests that AD pathogenesis is not limited to the neuronal compartment but comprises important interactions with immunological mechanisms in the CNS. It is now well accepted that neuroinflammation contributes to AD pathology and that this intricate immune response is not a passive system activated by emerging senile plaques and neurofibrillary tangles but instead contributes as much to pathogenesis as do plaques and tangles themselves [123,124]. Genetic studies support the crucial role of neuroinflammation since they disclose that genes for immune receptors, including TREM2 and CD33, are associated with a significant increase in the risk of AD [125,126].

Continually accumulating evidence is showing that neuroinflammation is a complex phenomenon and extremely variable from one subject to another, providing sometimes conflicting results. Indeed, preclinical and clinical evidence suggests the possibility that neuroinflammation wanes with age or during pathology progression. For instance, in 2011, some clinicians observed that the association between neuroinflammation and AD is stronger in relatively young patients than in the oldest patients [127], and more recently, other authors demonstrated that increased peripheral inflammation occurs early in AD—i.e., at the MCI stage—and decreases with the severity of the disease [128]. Further clinical studies also highlight the presence of inflammation since the early stages of the disease when clinical manifestations precede full-blown dementia, such as in MCI, supporting an early and substantial involvement of inflammation in AD pathogenesis [124].

Unfortunately, clinical trials that have been performed testing the potential of the most common anti-inflammatory drugs in AD have shown poor results [124], and anti-inflammatories have never entered clinical practice, which is still anchored to the use of drugs modulating the cholinergic and glutamatergic systems. The negative results of these clinical trials are likely related to not considering the positive effects exerted by neuroinflammation [93]. Indeed, neuroinflammation has both positive and negative aspects, and even if it can have detrimental consequences, it is basically a normal physiological response aimed at restoring homeostasis. Further investigations are needed to fully characterize both the positive and negative aspects of inflammation in AD and identify agents capable of counteracting the negative effects while preserving the positive ones.

## 8. The Role of Astrocytes, Microglia, and CNS Mast Cells in Alzheimer’s Disease

As stated above, neuroinflammation is driven by the activation of different brain cells, mainly microglia, astrocytes, and CNS mast cells. If the activity of these cells is under control, neuroinflammation operates correctly and then is turned off. On the contrary, if cell responses escape the physiological control systems, neuroinflammation remains “on” fostering pathological conditions, such as cognitive dysfunction and chronic pain (Figure 4).

It is now commonly believed that whenever a brain injury happens, the activation of glial cells takes place in order to remove the injurious stimuli. To this aim, both astrocytes and microglia undergo complex morpho-functional changes and acquire a so-called reactive phenotype (Figure 4) [129,130]. Reactivity causes glial hypertrophy, astrocyte endfeet retraction, and many other modifications that, if not stopped, can induce synaptic dysfunction, homeostatic imbalance, neurovascular unit dysfunction, loss of three-dimensional network, and blood-brain barrier (BBB) dysfunction [131,132]. In addition, reactive microglia and astrocytes release a wide range of pro-inflammatory mediators fostering inflammation [133].

Microglia modifications have also been described in humans, where the presence of microglial activation and proliferation in several brain regions of AD patients have been observed [134,135,136,137]. Very recently, a study in family-owned domestic dogs with CDS has shown a statistically significant increase of microglial numbers in CDS dogs compared to age-match controls [138]. Moreover, signs of microglia hypertrophy and activation were also observed [138].

Studies on animal models confirm astrocyte and microglia abnormalities in AD. For instance, genome-wide analysis suggests that several genes that increase the risk for late-onset AD encode factors that regulate the glial clearance of misfolded proteins and the inflammatory reaction [124]. Both microglia and astrocytes display pattern recognition receptors that bind misfolded and aggregated proteins and triggering an innate immune response characterized by the release of pro-inflammatory mediators responsible for disease progression [139,140].

Studies over the last twenty years have progressively brought to light the involvement of glial reactivity and inflammation in AD, even if the underpinning molecular mechanisms have not yet been fully clarified. For a long while, it was believed that glial cells acquire a reactive phenotype and foster inflammation upon β-amyloid deposition or neurodegeneration. Nowadays, accumulated evidence indicates that reactive gliosis and inflammation occur in the very early phase of the disease before histopathological modifications [127,128,141,142]. On the contrary, other studies on rodent models of AD indicate that the early astroglia response is represented by cell atrophy that may have important consequences for synaptic connectivity, thereby contributing to cognitive deficits [143,144,145,146,147]. These signs of atrophy appear first in the entorhinal cortex and affect astrocytes located afar from senile plaques in the later stages of AD.

In AD, it is possible to identify both glia reactivity and atrophy. These responses are considered pathological, and both contribute to the perturbation of brain homeostasis leading to neuronal damage and cell death. Consequently, neuroinflammation is not constantly present, but it is likely time- and context-dependent. Understanding the balance between the positive effects of glial reactivity and the negative effects is essential for understanding the pathogenic role of neuroinflammation in AD.

The localization of mast cells within the CNS has expanded the investigations on their possible roles in neuroinflammation and neurodegenerative disease, including AD [96,113,115]. CNS mast cells are now reputed as the primary first responders to damage by virtue of their secretory arsenal of stored proteases, growth factors, neuromodulators, and immunomodulators [94,148].

Some authors have hypothesized that mast cells are one of the first brain cells that sense amyloid peptides, thus exerting an important role in AD onset and (possibly) progression [149,150]. In agreement, post-mortem studies have shown mast cells surround amyloid plaques in AD patients in higher numbers in comparison to age-matched control patients [151]. The evidence available so far, although limited, supports a possible role for mast cells in AD pathology [105] and opens exciting new fields of research for neuroscientists.

Overall, mast cells, as well as microglia and astrocytes, are now considered critical effectors during several neuroinflammatory disorders, including AD [152]. The modulation of the crosstalk between mast cells and glial cells is emerging as a valuable approach to treat these brain pathologies [105,112,153].

## 9. Pro-Resolving Mediators in the Resolution of Age-Related Neuroinflammation

### 9.1. The Endocannabinoid System and Its Role in Brain Aging

Evidence accumulated in the last two decades indicates a prominent role of neuroinflammation in normal aging and neurodegenerative disorders. This has prompted scientists to find a way to counteract the negative aspects of neuroinflammatory processes. In this context, great emphasis has been given to endocannabinoid signaling, which controls (and is affected by) physiological aging and neuroinflammation [154].

In the early 1990s, the serendipitous identification of a G-protein coupled cannabinoid receptor (CB1), at which cannabinoid compounds are active in the brain, spawned an explosion in endocannabinoid research [155]. A second cannabinoid receptor (CB2) was identified and first cloned in 1993 [156]. Despite coupling to the same family of G proteins and sharing some ligands, CB1 and CB2 appear to differ significantly from one another in their signaling. Soon after, the endogenous cannabinoid ligands N-arachidonylethanolamine (anandamide, AEA) [157] and 2-arachidonoylglycerol (2-AG) [158] were discovered. These endogenous compounds were first discovered in the brain, but they are also present in the periphery.

The endocannabinoids are lipid signaling molecules formed from phospholipids precursors within cells; they are not stored in vesicles, but they are biosynthesized on-demand and undergo rapid metabolic deactivation [159]. Once released, endocannabinoids move in a retrograde direction and transiently suppress presynaptic neurotransmitter release by activating CB1 receptor-mediated inhibition of voltage-gated calcium channels [160]. The primary function of the endocannabinoids is to fine-tune neurotransmitter release, thus modulating synaptic efficacy and neuronal activity. Their wide distribution in the CNS suggests that endocannabinoids may play multiple functions, including the control of movement, memory and learning, nociception, reward processes, emotionality and anxiety, neurogenesis, and neuroinflammation [161]. Indeed, an upregulated endocannabinoid system has been observed in dogs with inflammatory CNS diseases, thus suggesting the endocannabinoid system might be a potential therapeutic target in neuroinflammation-associated disorders [162].

Animal and human studies indicate that the endocannabinoid system is importantly involved in aging, as demonstrated by the crucial changes that this system undergoes with old age. For instance, an old-age-dependent decrease in CB1 mRNA levels has been demonstrated in rodents [163], as well as in humans [164] and dogs [165,166]. Transgenic mouse models, in which selective components of endocannabinoid signaling were knocked out entirely or in specific cell populations, revealed that disrupting this system compromises fundamental aspects of the normal aging process. Mice with total deletion of CB1 show age-dependent acceleration of cognitive decline accompanied by faster loss of hippocampal CA1 and CA3 neurons compared to age-matched non-transgenic animals [167]. Moreover, these mice display an increased number of activated glial cells and augmented levels of pro-inflammatory cytokines [167]. Furthermore, mice lacking CB2 receptors present a phenotype that is remindful of accelerated aging, albeit outside of the CNS. These animals have severe osteoporosis [168], which is consistent with the suggested function of the CB2 receptor in bone biology. Altogether, these observations highlight that the disruption of cannabinoid receptors accelerates the age-related decline in many tissues and organs in which endocannabinoids have important physiological functions, supporting the idea that this system is crucially implicated in the control of the aging process [169].

### 9.2. The Endocannabinoidome

Since the discovery of AEA and 2-AG, it has been clear that these signaling molecules are accompanied in tissues by congeners. Further biochemical studies have allowed the identification of other molecules and led to the characterization of the so-called endocannabinoidome. The endocannabinoidome comprises several lipid ligands classified according to their chemical structures in N-acylethanolamines and monoacylglycerols. These molecules are further sub-classified in endocannabinoids (i.e., agonists at CB1 and CB2 cannabinoid receptors, such as the arachidonic acid derivatives AEA and 2-AG) and endocannabinoid congeners being inactive at cannabinoid receptors (such as PEA and several similar compounds not discussed in the present review [161]). Endocannabinoid congeners are biologically active and share with AEA and 2-AG their biosynthetic and inactivating enzymes, even if they do not activate CB1 and CB2 receptors. The brain concentrations of these congeners increase in animals treated with inhibitors of endocannabinoid hydrolyzing enzymes, as well as in animals lacking these enzymes [170,171]. These congeners activate several receptors highly expressed by different cell types of the CNS and are involved in aging, neuroinflammation, and neurodegeneration [172,173]. The most studied receptors are transient receptor potential vanilloid type 1 channel (TRPV1), peroxisome proliferator-activated receptor (PPAR)-γ and -α, and two orphan G-protein coupled receptors, GPR55 and GPR18 [174]. This expanded endocannabinoid system has been called the endocannabinoidome [161]. Figure 5 schematically describes the endocannabinoidome.

### 9.3. Endocannabinoidome and Neuroinflammation

Aging is accompanied by cellular modifications, such as changes in cellular processes, as well as by disruption of intercellular communication. At synapses, endocannabinoids provide a retrograde feedback system in which activation of presynaptic CB1 receptors reduces neurotransmitter release [175]. Moreover, endocannabinoids modulate the activity of glial cells, which may also be a source of brain endocannabinoids [176]. Increased activated astrocytes and microglia are typically detected in the aging brain and result in an augmented production of pro-inflammatory mediators. This process fosters a change towards a more pro-inflammatory milieu in the brain. Evidence accumulated so far indicates that endocannabinoids are generally protective against age-related pathologies, including neuroinflammation and neurodegeneration [174,177]. For instance, using several models of AD, it has been reported that AEA, cannabidiol, PEA, and other cannabinoid-related compounds attenuate the release of pro-inflammatory mediators from activated glial cells [142,173,178,179,180,181,182,183,184,185,186,187,188,189,190]. Moreover, cannabinoids inhibit oxidative stress, excitotoxicity, and calcium imbalance, all effects related to progressive neuronal death [191]. Many of these effects are mediated by the activation of CB1 receptors and provide the basis for potential therapeutic applications of cannabinoids in neurodegenerative diseases [174,192]. Moreover, the evidence showing that exogenous AEA exerts neuroprotective properties in models of excitotoxicity [193,194] in addition to the observation that mice with defective CB1 receptor gene are more vulnerable to neuronal damage [195] have further reinforced the hypothesis that the endocannabinoid system may represent a preservation system for the brain during neurotoxicity.

In the healthy brain, endocannabinoid production by neurons is high, and these cells express high levels of CB1 receptors in the dendritic tree and axon terminals, while the resting glial cells (mainly microglia) express low levels of CB2 receptors and produce low amounts of endocannabinoids [196]. During neurological diseases, in which the immune system has been activated, including AD, profound cell-specific modifications in the expression of cannabinoid receptors occur, resulting in lower levels of neuronal CB1 receptors and higher expression of CB2 receptors in reactive glia [197,198]. The latter result has been recently confirmed in dogs, with a strong upregulation of CB2 being observed in glial cells during inflammatory CNS diseases [162]. Changes in the levels of metabolic enzymes, lipid endocannabinoid congeners, and the expression of their receptors have been reported as well [182,184,197].

Considering the above, it is reasonable to hypothesize the development of new therapies for both symptom alleviation and disease modification in neuroinflammatory and neurodegenerative disorders based on the manipulation of endocannabinoidome signaling. This is undoubtedly a challenging but nevertheless fascinating task for neuroscientists to tackle in the future.

## 10. Pro-Resolving Mediators in the Resolution of Age-Related Neuroinflammation

Inflammation is a quite intricate network of cellular and molecular events with adaptive and defensive finalities aimed at repairing tissue injury and restoring homeostasis [199]. For this to happen, several stored or newly synthesized mediators are needed. Endogenous lipids are arguably the most important mediators implicated not only in the acute pro-inflammatory phase of inflammation but also in the regulation of its route and cessation [199]. During the resolution phase, the very same immune cells switch and start producing another class of bioactive lipids, called specialized pro-resolving mediators [200]. These compounds efficiently terminate inflammation and start the resolving phase [201].

### 10.1. Palmitoylethanolamide: An Endocannabinoid Congener Endowed with Promising Anti-Inflammatory and Neuroprotective Properties

Recently, endocannabinoids and their lipid congeners are also emerging as pro-resolving agents due to their ability to stimulate resolving programs during neuroinflammation [202,203,204]. Among these compounds, great attention had been focused on PEA, an AEA congener. PEA is an endogenous lipid compound, an amide of ethanolamide and palmitic acid, firstly isolated from soy lecithin [205].

For over a decade, this N-acylethanolamine was considered the progenitor of a class of substances (ALIAmides) that share the ability to appease the excess reactivity of non-neuronal and mast cells via the so-called Autacoid Local Injury Antagonism (ALIA) effect [205,206,207,208]. It is now well-established that PEA mimics several endocannabinoid-driven actions, even though it does not bind directly to cannabinoid receptors but other receptors belonging to the endocannabinoidome [205,209,210]. PEA has been detected in several regions of the spinal cord and brain, including the cortex, hippocampus, cerebellum, and thalamus/hypothalamus, of several mammals [211,212,213]. Remarkably, PEA is produced by either neurons, mast cells, or glia [182,214,215,216].

Interestingly, PEA is able to modulate the activity of both glial and mast cells, thus representing a very promising therapeutic tool to treat neurological disorders [217]. Several studies have indeed highlighted the ability of PEA to modulate the reactivity of these cells, mainly limiting the release of pro-inflammatory and neurotoxic mediators and favoring neuronal survival [180,181,218,219,220]. Two recently published studies confirm previous findings, as they show PEA to counteract reactive changes in microglia, reduce the expression of senescence markers and promote microglia polarization toward an anti-inflammatory phenotype [221,222].

These anti-inflammatory and neuroprotective effects are likely due to the ability of PEA to interact with the endocannabinoid system [223] and partially involve the CB2 receptor [222]. Under some circumstances, PEA may potentiate the actions of the canonical endocannabinoids, AEA and 2-AG, by increasing their levels [224]. PEA works as a false substrate for fatty acid amide hydrolase (FAAH), an enzyme involved in AEA metabolism [220]. In this way, due to the reduction of its catabolism, AEA levels increase. Thus, AEA binds to cannabinoid receptors. Moreover, PEA elevates 2-AG levels [224], whose concentration decreases with old age, especially in the hippocampus [225]. It has been proposed that PEA through the hippocampal elevations of 2-AG improves synaptic plasticity and mnemonic and cognitive performances [226]. In addition to these mechanisms, experiments using selective antagonists and murine models where PPAR-α was genetically ablated indicate that the anti-inflammatory and neuroprotective properties of PEA involve the activations of PPAR-α [180,210,227,228,229].

Through PPAR-α, PEA attenuated in vitro and in vivo β-amyloid-induced glial over-reactivity and the release of pro-inflammatory mediators, including inducible nitric oxide synthase (iNOS), cyclooxygenase (COX)-2, interleukin (IL)-1β, and tumor necrosis factor (TNF)-α [173,181,182,186,230,231]. PEA also promoted neuronal viability in primary astrocytes derived from the prefrontal cortex of 3xTg-AD mice, a transgenic murine model of AD [187]. All these data concur to demonstrate the key role of PPAR-α in controlling neuroprotective and anti-inflammatory pathways [232]. Moreover, studies showed that the protective effects exerted by PEA also include the activation of GPR55 [233] and TRPV1 [234]. Figure 6 summarizes the proposed mechanisms of action of PEA.

### 10.2. Dietary Supplementation with PEA-um as a Strategy to Control Age-Related Neuroinflammation and Neurobehavioral Correlates

PEA is considered a specialized pro-resolving mediator, and its role in mediating the inflammatory process has largely been demonstrated. It has been shown that under conditions of non-resolving neuroinflammation, such as occurring in AD, the endogenous levels of PEA in the CNS decrease [235]. A possible explanation could be the increased activity and expression of FAAH [235], which is one of its degradative enzymes or other mechanisms. Whatever way it happens, the reduction of endogenous PEA may lead to detrimental consequences as, for instance, insufficient concentrations of PEA in the hippocampus are related to memory deficits [236].

The reduction of endogenous levels of PEA during conditions characterized by non-resolving neuroinflammation, such as unsuccessful brain aging, suggests the opportunity to restore PEA endogenous reserves, thus fostering and re-activating physiological pro-resolving responses.

As PEA is a highly lipophilic compound, its bioavailability, especially after oral administration, is low [237]. This aspect limits the possibility of the use of PEA as such. However, in recent years, researchers have developed a technological modification that overcomes these difficulties because it reduces particles of PEA to infinitely small sizes, thus greatly improving its absorption by the oral route. This is the ultra-micronization process, which gives rise to the so-called ultramicronized (um)-PEA. After oral administration of um-PEA, the plasma concentration of PEA is five times higher than that measurable by administering non-micronized PEA [238]. After oral administration of um-PEA, PEA levels rise in the blood [239] and also in the brain [237], preferentially at the level of the hippocampus [240].

The pharmacological potential of um-PEA in controlling neuroinflammation—either singly or co-ultramicronized with the antioxidant luteolin (i.e., co-ultra PEA)—has been proven by numerous investigations in various models of neurodegenerative conditions and senile dementia, both in vitro and in vivo [241,242,243,244,245]. In rat hippocampal slices and neuroblastoma cells challenged with β-amyloid 1–42, PEA in the ultramicronized or co-ultramicronized formulation exhibited anti-inflammatory and anti-apoptotic effects, as well as the ability to decrease the expression of markers of oxidative stress and astroglial injuries, such as iNOS and GFAP [65,126]. In rats that received an intrahippocampal infusion of β-amyloid 1–42, chronic treatment for 14 days with co-ultra PEA prevented astrocyte hypertrophy as well as the production of pro-inflammatory cytokines and enzymes, compared to vehicle-treated animals [246]. Moreover, in this AD model, co-ultra PEA also prevented the decrease in gene expression of glial-derived and brain-derived neurotrophins [247]. In 3xTg-AD mice, i.e., an animal model of AD exhibiting age-dependent β-amyloid and tau pathologies, chronic administration of um-PEA reduced brain levels of several pro-inflammatory mediators and showed neuroprotective effects [142,187,240,248]. Importantly, um-PEA also prevented the impaired performance in cognitive tasks as well as reduced the AD-like pathology in these animals, as shown by the decrease of β-amyloid formation and tau protein phosphorylation in the hippocampus [142].

Collectively these results show that um-PEA acted towards normalizing glial activity and exerting neuroprotective effects, paving the way for clinical application. In human patients with frontotemporal dementia, a presenile neurodegenerative disease for which there is no effective pharmacological treatment, co-ultra PEA administration resulted in ameliorating behavior, cognition, and cortical activity through a combination of anti-inflammatory and neuroprotective effects [249].

All this evidence, together with the reassuring data on its safety and tolerability [244], strongly corroborates the use of um-PEA-containing formulations for the treatments of those conditions characterized by the presence of a long-lasting and non-resolving neuroinflammation, including aging, AD, and related dementias. These products are already available for both humans and pets, marketed as food for special medical purposes and complementary feed, respectively.

## 11. Conclusions

Brain aging in both physiological and pathological conditions is an intricate phenomenon and deserves much attention. Life expectancy increases, with more elderly dogs and cats being at risk of unsuccessful brain aging, from MCI to CDS. Therefore, it is mandatory to understand how to maintain brain health during the golden years. Growing evidence shows that aging and age-related disorders share some pathological trajectories, such as low-grade and non-resolving inflammation. The impact of this process on brain homeostasis is massive, causing detrimental consequences driven by neuroinflammation, such as impaired transmissions, altered behaviors, and cognitive dysfunction.

On the other side, it is necessary to keep in mind that agents able to counteract neuroinflammation should be effective and, more importantly, safe and devoid of serious side effects because treatment will possibly last for many years and even life-long. With this in mind, modulating the activity of endogenous pro-resolving systems might be a valuable option. In this context, the endocannabinoidome is a promising target, given its well-recognized functions in neuroinflammatory disorders. PEA belongs to this system and is an ideal candidate to manage age-related brain disorders, given a great deal of preclinical and clinical studies demonstrating its anti-inflammatory and neuroprotective functions as well as safety.

Increasing evidence suggests indeed that supplementation with um-PEA helps to accomplish successful brain aging, suggesting its dietary use as a new option for managing neuroinflammatory-driven cognitive dysfunction in aged animals.

## Figures and Tables

**Figure 1 animals-11-02584-f001:**
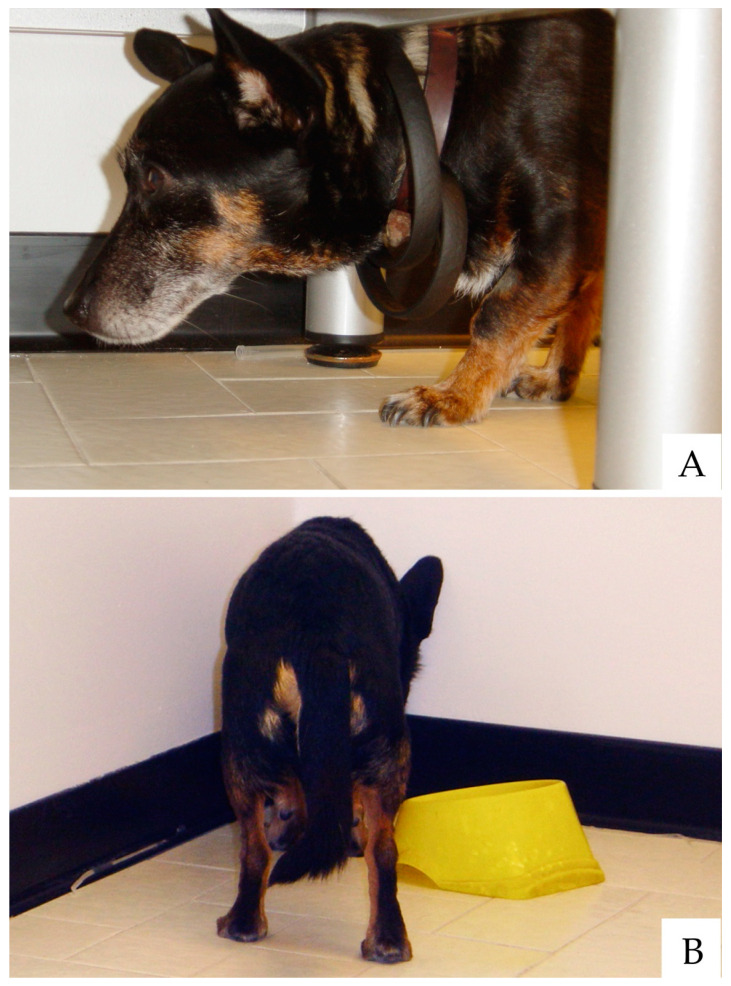
Old dogs often appear confused or disoriented. Getting “stuck” under furniture without apparently knowing how to get out from there (**A**) or standing headfirst in corners or tight spaces (**B**) are common signs of canine CDS.(photo by Lorenzo Golini)

**Figure 2 animals-11-02584-f002:**
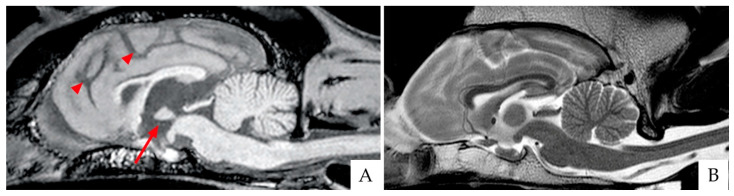
Brain scans of two breed-matched old dogs affected (**A**) and unaffected (**B**) by CDS, as acquired with 3 Tesla magnetic resonance imaging (3T MRI). (**A**) Sagittal T1 weighted image (T1WI). Please note the reduced size of the interthalamic adhesion (arrow) and the increased sulci (arrowheads). (**B**) Sagittal T2WI of the normal control dog.

**Figure 3 animals-11-02584-f003:**
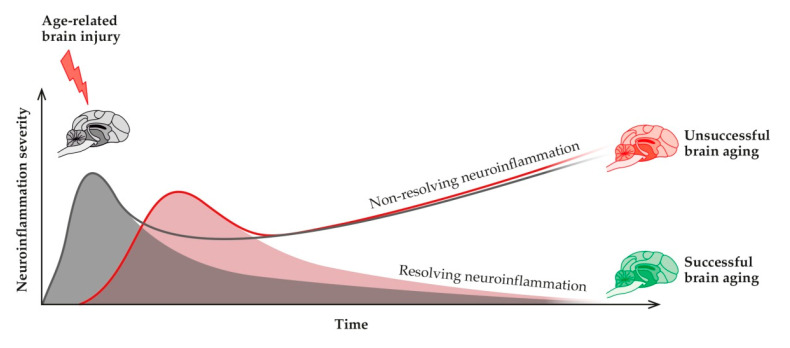
Neuroinflammation (red curve) is the physiological reparative response to age-related micro-injuries to the brain (grey curve). If the response is not correctly turned off, then non-resolving neuroinflammation might occur, with impaired adaptive brain responses, reduced resilience, and ultimately unsuccessful brain aging.

**Figure 4 animals-11-02584-f004:**
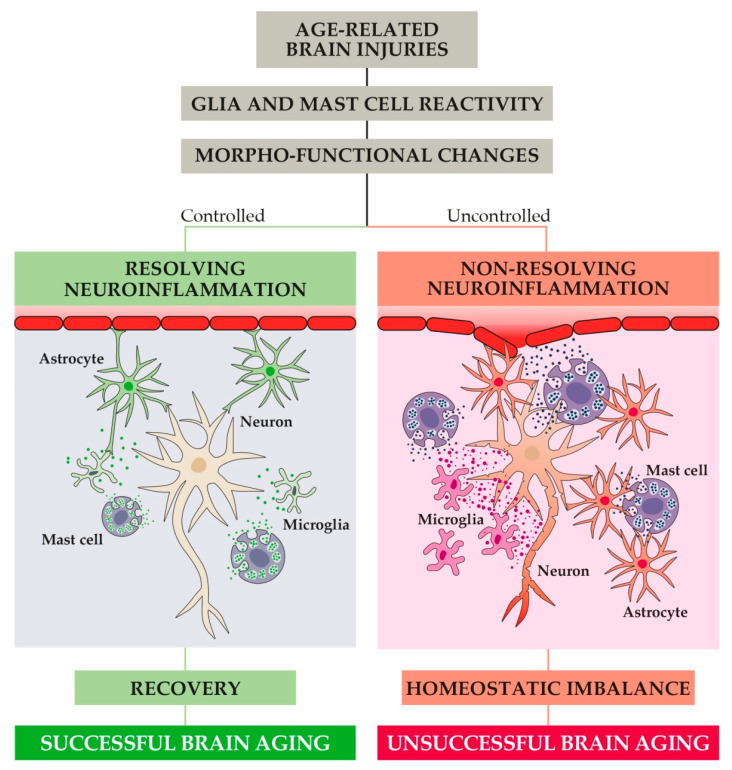
In response to age-related brain injuries, CNS immune cells (i.e., astrocytes, microglia, and mast cells) become activated and undergo profound morpho-functional changes toward a reactive phenotype. If the resulting release of pro-inflammatory mediators is not properly controlled, non-resolving neuroinflammation may begin. Under this condition, progressive dysfunction at the synaptic and neurovascular level occurs and ultimately leads to homeostatic imbalance and unsuccessful brain aging (i.e., impairments of cognitive and behavioral function).

**Figure 5 animals-11-02584-f005:**
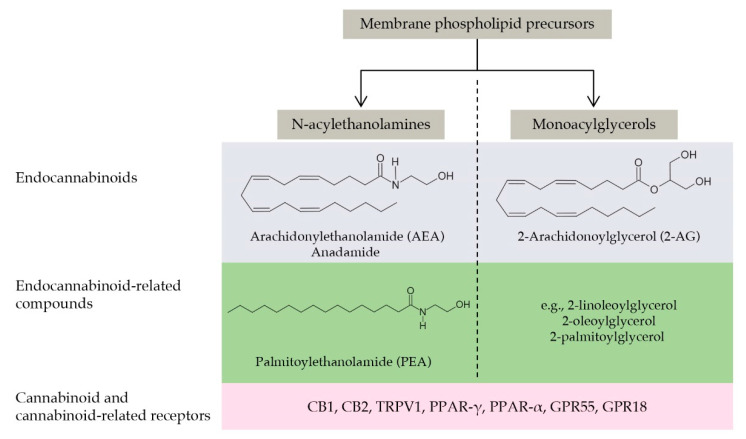
The main components of the endocannabinoidome: a schematic view. The endocannabinoidome comprises several lipid ligands classified according to their chemical structures in N-acylethanolamines and monoacylglycerols. Ligands are further subgrouped as endocannabinoids (i.e., agonists at CB1 and CB2 cannabinoid receptors) and endocannabinoid congeners being inactive at cannabinoid receptors. The arachidonic acid derivatives AEA and 2-AG belong to the first group, while PEA to the second one (together with several similar compounds not discussed in the present review). The endocannabinoidome also comprises the biosynthetic and degradative enzymes of these ligands (here not shown for the sake of simplicity) and the cannabinoid (i.e., CB1 and CB2) and cannabinoid-related receptors, such as those of the PPAR or GPR families. The reader is referred to Cristino et al., 2020 [174] for a more comprehensive review of this topic.

**Figure 6 animals-11-02584-f006:**
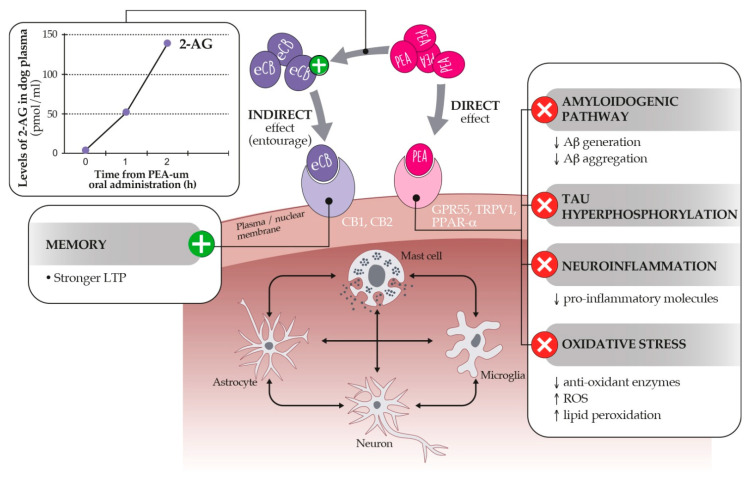
Proposed mechanisms of action of PEA. PEA exerts a direct and indirect agonism on the endocannabinoidome within brain cells. The direct pathway depends on the activation of GPR55, TRPV1, and PPAR-α receptors, the last inhibiting the hallmarks of age-related cognitive dysfunction (right panel). Through the so-called entourage effect, PEA may also increase the level (left panel) or the binding affinity of endocannabinoids for CB1 and CB2 receptors, favoring stronger LTP (Long Term Potentiation) and memory processes. See text for further details.

**Table 1 animals-11-02584-t001:** Main clinical signs associated with canine cognitive dysfunction syndrome. The behavioral and clinical categories are shown in bold.

**Mental status and spatial orientation (confusional status)** Get lost in a known environment Awaiting the door opening on the wrong side Inability to circumnavigate unknown objects Less interested in environmental stimuli
**Relationships (social interaction)** Less interested in being touched Ignoring the return of the owner Social behavior is disrupted Increased need for physical contact (is “needy”)
**Activity (increased—repetitive)** Starring at objects or empty space, fly biting Aimless walking Increased licking behavior (on the owner or objects) Increased vocalization
**Activity (diminished)** Apathetic, less interested in exploring Seems to not be interested anymore in known stimuli
**Appetite** Eats more than usual Eats less than usual
**Toileting behavior** Reduced time spent cleaning itself
**Anxiety (irritability)** Often irritable or anxious Shows signs of separation anxiety that has never had before Easily irritable
**Sleep—awake cycle** Short period of sleep interrupted by frequent abrupt awakenings Sleeps more than usual during daytime
**Learning and memory** Loss of housetraining, urinating or defecating in front of the owner Does not request to go out anymore Despite regular daily activity eliminates only when back home Eliminates where it sleepsIt is incontinent
**Learned behavior and commands** Struggle in performing a previously learned task Struggle to recognize a member of the family or other known people/animals Struggle to respond to commands Struggle to learn new commands or tasks

**Table 2 animals-11-02584-t002:** Main functions of CNS glia and mast cells.

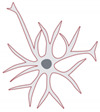 Astrocytes	-maintain CNS homeostasis at molecular, cellular, organ, and system levels [97].-regulate the communication between the CNS and the periphery (i.e., key components of the BBB) [98].-finely control the CNS microenvironment (e.g., extracellular ions, pH, blood flow) [99,100,101].-release about 200 molecules, mainly neurotrophic factors and energy substrates (gliocrine system) [102].-promote and finely control synaptic transmission, with an astrocyte being simultaneously in contact with several neurons (i.e., thousands of synapses) [95].
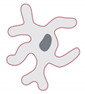 Microglia	-regularly scan the surrounding environment and restore homeostasis [103].-incorporate waste products, cellular debris, and pathogens (i.e., potentially phagocytic cells) [103].-fulfill defensive functions in the CNS, being the main immunocompetent cells of the brain [95].-secrete pro-inflammatory and anti-inflammatory molecules to direct the immune response in the CNS [104].-actively interact with astrocytes and CNS mast cells [105].-oversee the formation, shaping, and functioning of synapses [106,107].
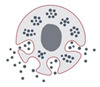 Mast cells	-mediate the interaction between the peripheral immune system and CNS during inflammation [108,109].-sense microenvironment changes and drive immune responses in the CNS [96].-establish reciprocal paracrine interaction with glial cells [109,110,111,112].-release several pro-inflammatory mediators, fostering neuroinflammation [113].-regulate BBB permeability through the release of proteases [114].-maintain BBB integrity through the release of vasoactive mediators (e.g., histamine and TNF-α) [115].

## Data Availability

No new data were created or analyzed in this study.

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
