# Peer review of "Successful and Unsuccessful Brain Aging in Pets: Pathophysiological Mechanisms behind Clinical Signs and Potential Benefits from Palmitoylethanolamide Nutritional Intervention"

_animals, 2021, doi:10.3390/ani11092584_

Round 1
Reviewer 1 Report
This manuscript has reviewed that dog aging and related behavioral changes and neuropathy. Then, it also discussed that the role of neuroinflammation in aging neuropathy. Finally, it described that the role of the endocannabinoid system in aging and neuroinflammation, and then the potential and mechanism of palmitoylethanolamide. Authors have collected and cited relevant documents, and properly planned the order of this manuscript and wrote them accordingly.
I have the following four suggestions:
- Please provide the list of abbreviations.
- Please provide the structures of palmitoylethanolamide, AEA, ECB, and 2-AG.
- Please explain the correlation among palmitoylethanolamide, AEA, ECB, and 2-AG.
- How does palmitoylethanolamide transform into AEA (or ECB) and 2-AG in the body
Author Response
1. Please provide the list of abbreviations.
Our reply: We thank the Reviewer for this suggestion. Accordingly, we included the list of abbreviations at the end of the manuscript
2. Please provide the structures of palmitoylethanolamide, AEA, ECB, and 2-AG.
Our reply: We thank the Reviewer for this suggestion. Accordingly, we included an additional figure (the new figure 5) depicting the endocannabinoidome in which the structures of some compounds are reported.
3. Please explain the correlation among palmitoylethanolamide, AEA, ECB, and 2-AG.
Our reply: We thank the Reviewer for giving us the possibility to clarify this important aspect. All these compounds are part of the so-called endocannabinoidome. We have edited paragraph 9.2 to highlight the correlation among these molecules.
4. How does palmitoylethanolamide transform into AEA (or ECB) and 2-AG in the body.
Our reply: We thank the Reviewer for giving us the possibility to also clarify this important aspect. Under some circumstances, PEA potentiates the activity of AEA and 2-AG. To the best of our knowledge, the conversion of PEA in AEA or 2-AG has not been reported yet. To make clearer this key point, we edited paragraph 10.1.
Reviewer 2 Report
In their review, Caterina Scuderi and Lorenzo Golini highlight various aspects of successful vs unsuccessful brain aging in pets and discuss the potential benefits of palmitoylethanolamide nutritional intervention. The authors performed an exhaustive search of existing literature. Overall, the review is well-planned. However, the following issues need to be resolved.
- While oxidative stress and inflammation more or less equally contribute to the pathobiology of brain aging, authors emphasize on inflammation. In fact, inflammation often leads to oxidative stress and vice-versa. Is there any particular reason to overlook oxidative stress?
- While there are several natural products having anti-neuroinflammatory potential, it is unclear why authors focus only on palmitoylethanolamide as an dietary intervention to brain aging in pets.
- Since this is a narrative review, the following recent articles deserve author’s attention.
- https://doi.org/10.1371/journal.pone.0238517
- 10.1007/s11357-021-00422-1
- doi: 10.3390/ijms21249526
Author Response
1. While oxidative stress and inflammation more or less equally contribute to the pathobiology of brain aging, authors emphasize on inflammation. In fact, inflammation often leads to oxidative stress and vice-versa. Is there any particular reason to overlook oxidative stress?
Our reply: We completely agree with the Reviewer on the fact that inflammation and oxidative stress contribute equally to the pathobiology of brain aging. Indeed, oxidative stress plays a key role in the loss of regulation of the neuroinflammatory response. A huge amount of data is available. In our opinion, it is really hard to discuss together these two key processes. This is why we decided to focus our efforts only on the role of inflammation in aging, which is a hot topic in neurology and more generally in neuroscience. Another reason we chose to focus on inflammation was our idea to report the evidence on the pharmacological effects of different formulations of PEA (a compound of which we have long experience, please see the following response) which is endowed with strong anti-inflammatory properties. We would like to thank the reviewer as it could be the subject of future works.
2. While there are several natural products having anti-neuroinflammatory potential, it is unclear why authors focus only on palmitoylethanolamide as an dietary intervention to brain aging in pets.
Our reply: Considering the amount of available data on this topic, this review was already quite extensive. We, therefore, decided to focus on the actions of a compound on which we have long experimental and practical expertise. For this reason, we decided to discuss exclusively different formulations containing PEA. This also correlates to the reason for which we have treated in a less incisive way the role of oxidative stress (please, see above). PEA indeed does not display antioxidant activity. However, we want to highlight, once again, that we fully agree with the Reviewer about the fundamental role that oxidative stress has in brain aging and its close correlation with inflammation.
3. Since this is a narrative review, the following recent articles deserve author’s attention.
- https://doi.org/10.1371/journal.pone.0238517
- 10.1007/s11357-021-00422-1
- doi: 10.3390/ijms21249526.
Our reply: According to her/his suggestion, in the revised version of the manuscript we have included the recommended articles (ref # 17, 86, and 245 respectively).